# Latest Insights and Therapeutic Advances in Myelodysplastic Neoplasms

**DOI:** 10.3390/cancers16081563

**Published:** 2024-04-19

**Authors:** Pasquale Niscola, Valentina Gianfelici, Marco Giovannini, Daniela Piccioni, Carla Mazzone, Paolo de Fabritiis

**Affiliations:** Division of Haematology, Sant’ Eugenio Hospital, 00144 Rome, Italy; valentina.gianfelici@aslroma2.it (V.G.); marco.giovannini@aslroma2.it (M.G.); pniscola@gmail.com (D.P.); carla.mazzone@aslroma2.it (C.M.); paolo.de.fabritiis@uniroma2.it (P.d.F.)

**Keywords:** myelodysplastic syndromes, eltrombopag, erythropoietin, molecular genetics, hypomethylating agents, luspatercept

## Abstract

**Simple Summary:**

Several recent studies have demonstrated encouraging results in treating patients with myelodysplastic syndromes/neoplasms (MDSs). This review focuses on the latest conceptual and therapeutic advances in the management of adults with MDS, addressing diagnostic principles, classification updates, and prognostic stratification systems, as well as improvements in clinical approaches that provide significant benefits through novel treatments, which may, in turn, represent a valuable basis for developing future clinical trials.

**Abstract:**

Myelodysplastic syndromes/neoplasms (MDSs) encompass a range of hematopoietic malignancies, commonly affecting elderly individuals. Molecular alterations in the hematopoietic stem cell compartment drive disease pathogenesis. Recent advancements in genomic profiling have provided valuable insights into the biological underpinnings of MDSs and have expanded therapeutic options, particularly for specific molecularly defined subgroups. This review highlights the diagnostic principles, classification updates, prognostic stratification systems, and novel treatments, which could inform future clinical trials and enhance the management of adult MDS patients, particularly for specific molecularly defined subgroups.

## 1. Introduction

Myelodysplastic syndromes/neoplasms (MDSs) [1] encompass a diverse group of hematopoietic malignancies [1,2], often observed in the elderly [3]. They are characterized by recurrent molecular mutations [4,5,6], affecting the hematopoietic stem cell (HSC) compartment [7,8], and are often preceded by clonal haematopoiesis (CH), such as idiopathic dysplasia of unknown significance, clonal haematopoiesis of indeterminate potential, idiopathic cytopenia of uncertain significance, and cytopenia of undetermined significance (CCUS). Therefore, CH may represent premalignant conditions, as revealed by genomic analyses [8,9,10,11]. Dysplasia in one or more cell lineages [1] and ineffective haematopoiesis are hallmarks of MDS, which the clinical phenotype is characterized by unexplained and persistent cytopenia (i.e., anaemia, neutropenia, and/or thrombocytopenia) due to abnormal cellular maturation. Therefore, patients are at risk for symptoms related to anaemia, infection, and bleeding, as well as for progression to acute myeloid leukaemia (AML) [6,12]. In the present review, we aim to explore the latest insights concerning the diagnostic principles and prognostic and risk stratification systems, as well as updated classifications and therapeutic advances in MDS management in adults.

## 2. Search Strategy and Selection Criteria

References for this updated review were identified through searches of PubMed with multiple search terms related to several aspects of biology, diagnosis, prognostication, and clinical management of adult individuals with MDS. Only papers published in English until March 2024 were reviewed. With some limited exceptions, only papers published over the last three years were considered with the aim of providing a summary as up to date as possible of the most recent developments on the topic. The final reference list was generated based on originality, reproducibility, and relevance to the scope of this review.

## 3. Disease Overview and Pathogenesis

The clinical onset of MDS is typically subtle, being related to the worsening of one or more cytopenias, including anaemia, thrombocytopenia, and less commonly, neutropenia [12]. Asymptomatic patients are often identified through routine blood tests, while others may present with fatigue [13] and general malaise, prompting these symptoms for further haematological evaluation. These cytopenias may arise from long-lasting CHs, which are prevalent in individuals over 70 years of age [3,11], being typically associated with aging and, sometimes, with immune dysfunctions. Aging hematopoietic stem cells (HSCs) exhibit pathological changes, including immunological alterations [14], such as pre-existing autoimmune diseases that may impact 10% to 30% of MDS patients [14], as well as abnormal activation of inflammatory pathways [8,15]. Indeed, studies conducted during the past few decades have highlighted the complexity of MDS pathophysiology involving the genetic diversity of clones, inflammatory conditions related to aging, and the impaired response of the immune environment [7,8], where HCSs may be involved by recurrent cytogenetic aberrations, such as 5q deletion (del5q), deletion or monosomy 7, 20q deletion, trisomy 8, as well as genetic mutations affecting epigenetic regulators (*TET2*, *IDH1/2*, *DNMT3A*, *ASXL1*, *EZH2*), splicing factors (*SF3B1*, *SRSF2*, *U2AF1*, *ZRSR2*), transcription factors (*TP53*, *RUNX1*), signalling adapters (*NRAS*, *KRAS*, *JAK2*, *CBL*, *KIT*), and cohesins (*STAG2*, *SMC1*, *SMC3*), as revealed through next-generation sequencing (NGS) [4,5]. Therefore, common genomic lesions in MDS involve loss-of-function mutations in some genes, such as *DNMT3A* or *TET2*, leading to increased expression of inflammatory signalling genes and the release of pro-inflammatory molecules [8,10]. The interplay between immune dysfunction and molecular mutations plays a pivotal role in clonal expansion and, disease progression [8,11,14,15]. Moreover, inflammatory mediators can disrupt the BM microenvironment and HSC niches [7,8], fostering the selective growth of abnormal blood cells with genomic mutations while suppressing normal haematopoiesis (Figure 1). Immunological pathogenetic mechanisms have been recognized in the development of the hypoplastic MDS (h-MDS) [1,7,16]. Indeed, this MDS disease entity is associated with a T-cell-mediated immune attack on HSCs [15,16] along with the oligoclonal expansion of CD8 and other cytotoxic T-cells. Moreover, increased production of pathogenic interferon (IFN)-γ and/or tumour necrosis factor (TNF)-α has also been reported in this particular setting [16]. Again, another form of MDS concomitant with an immunological disorder is diagnosed in two-thirds of patients affected by vacuoles, E1 enzyme, X-linked, autoinflammatory, somatic (VEXAS) syndrome. The latter is a prototypic haemato-inflammatory disease combining rheumatologic and hematologic disorders in a new disease entity, which is molecularly defined by somatic mutations in the UBA1 gene [17,18,19]. Furthermore, molecular and sequencing advances have led to the discovery of new genes and inherited genetic abnormalities (germline mutations) associated with an increased risk of developing myeloid malignancies, including MDS [20,21,22]. In addition, a remarkable proportion of MDSs are therapy-related (t-MDS) and can emerge as a late consequence of prior chemotherapy or radiotherapy exposure for a previous non-myeloid neoplasm [23,24,25]. In this regard, the occurrence of t-MDS in patients with lymphoid malignancies has recently been reported in the innovative rescue setting of Anti-CD19 Chimeric Antigen Receptor T-cell Therapy (CAR-T) [26]. Notably, none of the patients had dysplastic clones before the initiation of CAR-T, and two cases of MDS were diagnosed at 10 and 26 months after. Moreover, two cases of CCUS developed one and two months after CAR-T [26]. The new 2002 International Consensus Classification (ICC) [2] eliminated t-MDS, and therapy-related cases are now subclassified following primary diagnosis [2]. Conversely, the updated Fifth World Health Organization Classification of Myeloid Neoplasms (WHO-5) [1] continues to distinguish them as a specific disease entity. However, it is essential to recognize that secondary diseases carry a worse outcome, being frequently associated with poor prognosis TP53 mutations and complex karyotypes, as well as a shorter median survival compared to current treatment strategies [23,27,28].

## 4. Diagnosis and Classifications

MDS diagnosis traditionally relies on clinical and morphologic characterization, as well as genetic criteria [29,30]. The recognition of dysplastic features is crucial for MDS diagnosis, with the recommended threshold for dysplasia set at 10% for all lineages [2]. Furthermore, CBC counts, myeloblast percentage [30,31,32,33], and karyotype [4,34] have traditionally represented the main clinical and pathological variables contributing to risk stratification [35,36] and allowed for the recognition of some MDS-specific subtypes [16,37,38,39,40]. Moreover, the clinical significance of BM cellularity and reticulin fibrosis degree [32,33] by immunohistochemistry are now recognized by WHO-5 [1] and the 2022 ICC [2]. Furthermore, immunohistochemistry can be useful for identifying the cellular lineage and/or the aberrations of cellular maturation [33]. Indeed, staining for glycophorin (CD235a), transferrin receptor (CD71), and/or GATA1 can aid in detecting erythroid precursor cells, whereas immature or dysplastic megakaryocytes can be revealed by CD41 and/or CD61 analysis. Again, staining for myeloid and lymphoid markers can help to detect lineage infidelity, confirming or excluding the presence of bi- or tri-lineage dysplasia as well as detecting the origin of primitive blasts and progenitors. In this regard, CD34, CD117, CD33, myeloperoxidase, and lysozyme staining can assist in quantifying myeloid blasts [33]. In addition, a useful accuracy in blast counting, other than valuable findings on the pathological immunophenotype features of MDS cell populations, can be provided by multiparametric flow cytometry (MFC), although this methodology has still only a complementary role [29] in this field and it is not yet widely used in MDS diagnosis [41]. Although cytogenetic analysis maintains a fundamental role in the diagnosis and especially prognosis of MDS patients [35,36], with about half of them carrying one karyotype alteration [4,34], recent genomic advances have provided remarkable improvement in this setting [5,42]. Genetic advance have demonstrated at least one oncogenic genomic alteration in 94% of patients with MDS and have lead to the formulation of the Molecular International Prognostic Scoring System (IPSS-M) [42], in which development diagnostic samples from 2957 patients with MDS were profiled for mutations in 152 genes implicated in myeloid neoplasms [42]. The IPSS-M provided a focused list of 31 genes that can be considered for clinical practice as a minimal diagnostic gene panel that should be given to every MDS patient for proper evaluation and treatment decision making. Moreover, other commercially available panels have been reported in clinical practice, such as one exploring the mutational status of 69 potentially oncogenic alterations, including specific hotspots, full transcripts, and fusion transcript genes [43]. The recent advances in molecular genetic insights [5,43,44] have provided new diagnostic tools and prompted two novel classifications, i.e., the WHO-5 [1] and the ICC [2]. The WHO-5 classification introduces the term “myelodysplastic neoplasms” to replace “myelodysplastic syndromes”, underscoring their neoplastic nature. Moreover, it subdivides the MDS subtypes as having “defined genetic abnormalities” and those that are “morphologically defined”. Furthermore, the WHO-5 classification distinguishes, as an additional modification, between MDSs with low blasts (MDS-LBs) and those with increased blasts (MDS-IBs). Although the 20% blasts cut-off to delineate the boundary of MDS-IB2 and AML was retained, a large agreement suggests considering MDS-IB as AML-equivalent for therapeutic considerations and from a clinical trial design perspective [2]. Concerning defining genetic abnormalities, the WHO-5 classification recognizes three distinct groups, such as MDSs with low blasts and isolated 5q deletion (MDS-5q) and MDSs defined by a molecular alteration [1,38], such as *SF3B1* mutations (MDS-SF3B1) [1,39,40] and biallelic *TP53* inactivation (MDS-biTP53) [1,45]. However, for therapeutic considerations and in the light of published data [46,47,48], the latter disease subtype may be considered as AML-equivalent [1]. The same three specifically genetic-related MDS subtypes (MDS-5q, MDS-SF3B1, and MDS-biTP53) recognized by WHO-5 are also reported in the ICC classification [2]. However, in the latter, MDSs with excess blasts (MDS-EBs) were defined by the presence of at least 5% myeloid blasts in the BM or at least 2% blasts in the peripheral blood (PB). In addition, the blast threshold of 20% defining AML was maintained but several additional genetic founding lesions were considered to be defining of AML for MDSs with ≥10% BM or PB blasts. Again, to underscore the biological continuum between MDSs and AML, the previous category of MDS-EB2 in adults with 10% or more blasts was changed to MDS/AML [2,31]. The ICC and WHO-5 classifications have important implications for medical practice and clinical research in this field [49,50,51,52,53,54,55]. Notably, both classification systems share more similarities than differences, demonstrating their value in recognizing MDS distinct entities, some of them suitable for tailored therapies [1,2,38,49], and specific prognostic risk groups [52,53,54]. Moreover, the previously defined categories of MDSs with excess blasts have been refined. Indeed, MDSs with a higher blast percentage are more likely to transform into AML, carrying a worse prognosis similar to overt AML [31]. Therefore, a new entity “MDS/AML” to define MDSs with blasts between 10 and 19% and to more accurately reflect the continuum between MDSs and AML was suggested [2], conceptually expanding AML treatment modalities or clinical trials for this new category of patients with MDS/AML [31]. From both criteria, most patients (70%) were classified morphologically, while the remaining 30% were genetically defined [54]. Furthermore, the incorporation of features from the revised International Prognostic Scoring System (IPSS-R) [56] and molecular IPSS (IPSS-M) [42,57] provided additive prognostic and survival components to both classifications [54]. Therefore, the WHO-5 and ICC classifications represent significant advancements in the diagnosis and classification of MDS, providing valuable prognostic information and guiding clinical management decisions. Further efforts to merge them to develop a re-unified MDS classification could facilitate clinical research and improve the comparability of clinical trials in MDS [55]. Table 1 reports a comparison between the WHO-5 and ICC 2022 classifications, highlighting their differences and updates in terms of blast percentage categorization and the inclusion of genetic abnormalities and morphological features [1,2].

## 5. Prognostic Systems and Risk-Stratification

Effective risk stratification is crucial in MDS management, guiding therapy selection based on the risk of transformation into AML. In this regard, MDS patients are categorized into lower-risk (LR) and higher-risk (HR) groups, with expected survival ranging from a few months to over 10 years [12,35,36]. Traditional clinical prognostic scoring systems, such as the International Prognostic Scoring System (IPSS) and its revised version, IPSS-R (Table 2) [56], have been widely used in medical practice and clinical research. However, integrating mutational data through NGS led to the development of the IPSS-M [42,57]. At least one somatic mutation is found in more than 90% of MDS patients [42,43,44,45]. Although clonal evolution, with the acquisition of other oncogenic lesions, characterizes the progression from LR to HR MDS, the NGS panel for targeted sequencing shows some prevalently mutated genes in the distinct prognostic group [52,53]. In particular, in a study including 366 MDS patients, 20 genes were mutated in almost one-third of HR and very HR R-IPSS patients, with the most prevalent being *KDM6A*, *CEBPA*, *TP53*, *ASXL1*, *RUNX1*, *DNMT3A*, *SH2B3*, and *BCOR* [50]. In LR categories, the most affected mutated genes, although at a lower prevalence, are *TET2*, *ASXL1*, *SF3B1*, and *U2AF1* [52]. In addition, in a recently report [53] on 79 MDS patients showing +8 isolated cytogenetic alterations, the mutational profile identified an HR subgroup with mutations in *STAG2*, *SRSF2*, and/or *RUNX1*, resulting in independent prognostic factors of a shorter time to AML evolution and overall survival (OS). Moreover, 39.5% and 15.4% of patients classified as low/intermediate risk by the IPSS-R and IPSS-M, respectively, were re-stratified as an HR group based on the mutational status of *STAG2*, *SRSF2*, and *RUNX1* [53]. So, incorporating molecular data improved risk identification, with a significant proportion of patients moving into the HR MDS groups, facilitating personalized treatment decisions and identification of therapeutic targets. In addition, and as above outlined, patient-reported fatigue is a major subjective complaint reported by individuals with newly diagnosed MDS [13]. Moreover, in HR-MDS, the severity of this symptom at diagnostic workup was clinically meaningful and provided prognostic value [58,59,60]. On this basis, a novel patient-centred prognostic index that includes patients’ self-reported fatigue severity, such as the fatigue IPSS high risk (FA-IPSS(h)), was proposed by the authors, claiming that it might enhance a physician’s ability to predict overall survival (OS) more accurately in patients with advanced MDS [58]. Again, the same working group reported that compared with the IPSS classification, concerning fatigue severity, the IPSS-R provided a better stratification of patients [60]. More recently, compared to IPSS-R, a higher prognostic accuracy in survival prediction was achieved for MDS patients using the Artificial Intelligence Prognostic Scoring System, which takes into account exclusively traditional clinical, haematological, and cytogenetic parameters [61].

## 6. Clinical Management

Nowadays, effective clinical management hinges on risk stratification, based on a comprehensive workup, considering clinical variables, such as cytopenias, transfusion needs, and BM/PB blast infiltration other than the fundamental cytogenetic findings and mutational profiles [62,63]. In addition, quality of life (QoL) [64] has an ever-growing importance in clinical practice and research. In addition, the evaluation of comorbidities has a crucial role in decision making [65] and in establishing potential eligibility for allogeneic stem cell transplantation (SCT) [66,67,68,69,70]. Generally speaking, MDS patients with an LR of AML evolution require treatment of their cytopenias, while patients with HR-MDS are expected to undergo pre-emptive therapies of disease progression and curative treatments whenever applicable [62,63]. 

Active clinical surveillance is required given the propensity of MDS to progress into more aggressive disease compared to AML. Particularly attention, in clinical practice as well as in counselling of family members, must be given to MDS patients with germline mutations [20,21,22]. Regardless of the MDS subtype and prognostic allocation, anaemia is prevalent in 80% of MDS patients, and 50% of them receive red blood cell (RBC) transfusions during their disease [71,72,73]. Of note, anaemia necessitates focused management given its significant impact on QoL [64] and organ damage, which may be secondary to demand ischaemia, increased cardiac output, and left ventricular remodelling [74,75]. In addition, anaemia exerts other specific adverse effects, such as immunological reactions and healthcare costs, as well as health concerns, such as iron overload [75,76,77]. Moreover, transfusion dependence may be very harmful or burdensome to patients and families [64,78]. This section aims to address the current practice and next potential advances in the clinical management of LR- [43,79] and HR-MDS [80,81]. The former MDS group includes some specific MDS subtypes, such as MDS-5q [1,38], MDSs with low blasts, MDS-SF3B1 [39,40,49], ad h-MDSs with a low blast count [16], which are traditionally considered among LR-MDSs in the clinical praxis, although they are now recognized as distinct disease entities by novel classifications [1,2]. Furthermore, MDS-biTP53 entities [45,46,47,48], in light of the new classifications [1,2] and its unfavourable clinical course and poor prognosis, are addressed at the end of the paragraph that deals with HR-MDS. Lastly, the crucial topic of allogeneic SCT [66,67,68,69,70], the fundamental goal for eligible patients and the only real treatment opportunity in this difficult-to-treat setting, will be addressed in a distinct paragraph. 

## 7. Lower-Risk MDS

The clinical management of LR-MDS focuses on treating symptoms and cytopenias, among which anaemia is the hallmark of the disease and the most common indication for treatment [43]; however, isolated thrombocytopenia and neutropenia are encountered less often [62,82]. Therefore, the main therapeutic efforts are aimed at correcting chronic anaemia [71,72,73] and thrombocytopenia [83,84,85], reducing recurrent infections [82], and improving or maintaining QoL [58,64,86]. Figure 2 shows an updated algorithm proposed for the treatment of LR-MDS [62]. Despite the latest therapeutic advances, LR-MDS patients are largely managed with supportive care, including RBC transfusions and erythropoiesis-stimulating agents (ESAs), which include recombinant humanized erythropoietin (EPO) agents or the longer-acting EPO (darbepoetin alfa), representing the initial therapy of anaemia [62,63]. In this regard, ESAs can improve anaemia in 40–70% of LR-MDS patients with low or no transfusion dependence; responses can be sustained, with a median duration of 12–18 months [87,88]. Other than traditionally recognized predictive factors for ESA response, such as the low basal level of serum EPO (<500 mU/mL) and no need for transfusion [63,87], some others have been recently identified. Indeed, the adverse effect of somatic mutations, which have been associated with prognosis and response to ESAs, has been reported in LR-MDS patients [72]. In particular, a higher frequency of erythroid responses among patients with a lower (<3) number of some mutated genes was reported. Conversely, a worse OS and a higher cumulative AML progression were observed in patients with ≥3 mutated genes, even in ESA responders [72]. Therefore, an impaired response to EPO underlies ineffective erythropoiesis with anaemia exacerbation and transfusion needed in LR-MDS patients, limiting the available options for those refractory to or relapsed (R/R) after ESAs [71,88]. Some remarkable exceptions are represented by del(5q) MDS, responsive to immunomodulatory drugs, notably lenalidomide [89], and MDS-SF3B1, which can be treated with luspatercept [90]. However, those with h-MDS with a low blast count may receive immunosuppressive treatments in selected cases [43,62]. Outside these specific disease entities, the treatment approach is based on RBC transfusion alone and the inclusion in clinical trials whenever available. In this regard, some attempts have been made to reverse EPO-failed anaemia in patients, for which no specifically targeted treatments are available. The potential rescue role of lenalidomide in EPO-failed non-del (5q) MDS patients has been explored, reporting in [91] the achievement of transfusion independence longer than 8 weeks in 26.9% of patients, achieving the response within 16 weeks of treatment [91]. Notably, these responses impacted OS by the reduction of transfusion burden and ferritin levels [92]. Furthermore, in the lenalidomide-treated patients, the presence of a mutation in any five gene, such as *ASXL1*, *ETV6*, *EZH2*, *RUNX1*, and *TP53*, was associated with a significantly shorter median OS, whereas those involving *SF3B1*, *TET2*, and *DNMT3A* had no significant effect on the patient’s outcome [93]. Again, a shorter leukaemia-free survival and a worse OS in lenalidomide-treated patients carrying an increased mutation burden (≥four mutations) were also reported [93]. The addition of EPO to lenalidomide in anaemic EPO-failed non-del (5q) MDS patients yielded an erythroid overall response rate (ORR) of 46.5% in patients who received the combined treatment compared to 32.3% for the lenalidomide monotherapy arm [94]. In addition, responses to the combination therapy were highly durable with a median duration of 23.8 months compared with 13 months reported in patients who received lenalidomide alone [94]. The authors concluded that lenalidomide, providing augmented EPO receptor signalling in vitro, can restore and improve haemoglobin responses to EPO in patients with LR non-del(5q) MDS who have anaemia that is refractory to or have a low probability of benefit from treatment with recombinant EPO [94]. In another study, predictive factors of a response to lenalidomide in non-del (5q) MDS, such as a low percentage of BM lymphocytes and progenitor B-cells, a low number of mutations, and the absence of RS and *SF3B1* mutations, were identified through MFC and NGS [95]. 

In EPO-failed RS-MDS patients, the outcome was poor until the recent availability of transforming growth factor β superfamily ligand inhibitors, such as luspatercept [90], which promote late-stage erythroid maturation to provide significant clinical benefits and QoL improvements [96]. In particular, patients not transfused and those with baseline low transfusion burden were identified as having a distinct probability of having a clinical benefit from the treatment [90]. Therefore, based on the clinical benefits provided by luspatercept, a clinical trial compared this agent with epoetin alfa for the treatment of anaemia due to LR-MDS in ESA-naive patients [97]. In this report, luspatercept improved the rate at which RBC transfusion independence and increased haemoglobin were achieved compared with epoetin alfa, including non-mutated *SF3B1* and RS-negative subgroups [90]. Furthermore, another promising compound, the telomerase inhibitor imetelstat [98], is in advanced clinical development, showing encouraging results in ESA-failed patients, for which approval in clinical practice is highly awaited. As rescue measures, the use of hypomethylating agents (HMAs), which are not approved in Europe for LR-MDS, has been reported [62,63,99]. In such a setting, adults with low- or intermediate-risk MDS or CML were randomized to receive either low-dose decitabine or azacytidine. With 68 months of follow-up, 67% and 48% of patients achieved significant disease responses in the decitabine and azacytidine arms, respectively [99]. Therefore, HMAs may be a suitable therapeutic option in selected LR-MDS patients in need of salvage therapy. Therefore, treatment strategies in LR-MDS patients firstly aim to manage anaemia to avoid or reduce the need for transfusion [43,63,79], as well as prevent some clinical complications, in particular, deleterious effects related to iron overload [76,77], which should be managed with iron chelation therapy with deferasirox. Iron chelation has been proven to provide clinically meaningful improvements [100,101] and should be introduced in the treatment plan of transfused patients as soon as possible, taking into account the regulatory system of each country. 

Other than anaemia, managing other cytopenias [43,69] may represent a challenging concern, although some novelties can change this scenario [85]. Indeed, severe thrombocytopenia affects 10% of MDS patients and is associated with poor outcomes. However, its optimal management, traditionally based on prophylactic platelet transfusions, is not well established [75]. Disparities in clinical practice are likely related to MDS individuals and provider heterogeneities [83,84], for which the majority of thrombocytopenic patients receive platelet transfusions, although only a tenth of them develop major bleeding, occurring at a wide range of thrombocyte counts [84]. The only available treatment modalities that can potentially improve platelet counts are immunosuppressive therapy; androgen agents, such as danazol [102]; and HMAs [63,99], with the latter agents not approved in all countries for LR-MDS. However, as reported by a recently published multicentre trial, eltrombopag was effective and relatively safe in LR-MDS patients with severe thrombocytopenia [85]. Lastly, no specific treatment has been established for MDS-related neutropenia, and the role of granulocyte colony-stimulating factor in this setting is mostly reserved for febrile neutropenia [62,82].

## 8. Higher-Risk MDS

Compared to LR-MDS, the clinical management of HR patients significantly differs, in line with the AML approach [2,62,63], aiming at modifying the natural disease history, prolonging survival [80,81], and possibly achieving adequate disease control or cure through allogenic STC, which should be considered early in the disease course and promptly proposed to eligible patients [66,67,68]. HMAs, given with a life-prolonging intent and/or to reduce disease burden for subsequent allogeneic SCT, are still currently the standard of care for patients with HR-MDS [62,63,103,104,105]. Figure 3 reports the current clinical practice of difficult-to-treat patients [62]. However, only around half of patients respond to these agents. In addition, responses are transient and generally last less than two years [104,105]. Thus far, several novel agents with different mechanisms of action have been tested, mostly in association with HMAs, to improve the treatment efficacy in HR-MDS [103]. Indeed, the increasing knowledge of MDS pathogenesis has led to the development of new potential therapies. In particular, HR-MDS presents an abnormal overexpression of the antiapoptotic BCL-2 protein, which is associated with the maintenance and survival of blast cells, treatment resistance, and poor OS. Thus far, specific inhibition of its activity and the activation of the endogenous mitochondrial apoptotic pathway by venetoclax causes rapid tumour cell death [106]. Therefore, emerging treatments, such as venetoclax combined with azacytidine, are transforming HR-MDS management, potentially leading to a paradigm shift [62,106,107]. Indeed, this treatment combination was investigated de novo in HR-MDS patients, achieving an ORR of 74% [107]. Myelosuppression, particularly neutropenia, was the most reported adverse effect. Therefore, the addition of venetoclax to azacytidine may benefit HR-MDS patients, formerly with AML/MDS according to the ICC classification [2], with two-thirds of them achieving complete remission (CR) or CR with incomplete peripheral recovery (CRi) [107]. The high efficacy of the venetoclax–HMA combinations in this setting has prompted phase III clinical trials. These treatment combinations may hopefully become available very soon in clinical practice. Currently, there are no standard-of-care options for HMAs–R/R patients, for which the disease progression portrays a poor prognosis with a very dismal median OS [108]. In addition, outside clinical trials, in which these patients should be recruited [95,100], very limited therapeutic options are available, taking into account that intensive chemotherapy and allogeneic SCT are only feasible in a minority of selected cases [62,63]. The safety and efficacy of the venetoclax/azacytidine combination were tested in this daunting clinical challenge in a phase I study [109], including 21 patients with R/R HR-MDS, 39% of which achieved a response with a median time to CR (7%) or CRi (32%) of 1.2 months. In addition, transfusion independence for RBCs and/or platelets was achieved in 36% and 43% of patients, respectively. Of note, with a median follow-up of 21.2 months, the median response (CR and CRi) duration was 8.6 months. Again, the median OS of responding patients was 12.6 months [109]. Again, an ORR as high as 57% and a median OS of 14 months were reported by a retrospective study dealing with the real-world treatment of R/R MDS patients with azacytidine plus venetoclax, the latter agent given for a limited course (15 days) for each treatment cycle to limit the myelosuppression and avoid infectious complications [110]. Therefore, the combination of venetoclax with azacytidine demonstrated high efficacy, offering hope for HR-MDS patients with limited treatment options [110], and its efficacy and safety were also reported in difficult-to-treat relapsed MDS post-allogeneic SCT [111]. A recently published study investigated the cytogenetic and molecular alterations that may help identify which HR-MDS patients may benefit the most from venetoclax. In this regard, prior HMA failure, complex cytogenetics, trisomy 8, *TP53* mutation, and *RAS* pathway mutation were all associated with inferior outcomes, whereas molecular alterations in RNA splicing, DNA methylation, and *ASXL1* were favourable. Interestingly, blast percentage was not predictive of outcomes [112]. Other than venetoclax-based treatments, other measures, such as newer formulations of intensive chemotherapy, e.g., CPX-351, may be considered a valid option in selected patients after HMA failure [113], as was reported by a recently published paper reporting an ORR of 56% and a median relapse-free and OS of 9.2 and 8.7 months in HMA-failed patients with HR-MDS and chronic myelomonocytic leukaemia, respectively [113]. The clinical outcomes in patients with MDS-biTP53 [46,47,48] are poor and marked by HR clinical features, such as complex karyotype, prior exposure to leukaemogenic antineoplastic agents, low ORR to HMAs [46], and short survival after allogeneic SCT due to elevated risks of relapse [46]. So, for therapeutic considerations, this MDS subtype should be considered as AML-equivalent [2]. However, novel therapeutic approaches may be offered by recently developed targeted agents, such as eprenetapopt, which is a novel, first-in-class, small molecule that restores wild-type p53 functions in TP53-mutant cells [114]. This agent has been combined with the backbone azacytidine [115] plus venetoclax in a triple combination [116]. These combined treatments including eprenetapopt, resulted in well-tolerated yielding high rates of ORR including molecular remissions in this difficult-to-treat setting. These favourable results may open new ways to treat these poor-prognosis diseases, offering opportunities to modify the approach to pre-transplant conditioning or post-transplant maintenance and improve clinical outcomes [46,47,48]. 

## 9. Allogenic SCT

Allogeneic SCT is the sole curative therapy for MDS patients [12,63]. It is recommended to evaluate the potential for this procedure at the onset of HR-MDS and even in patients initially diagnosed with LR disease [62,79] who require close clinical surveillance. Regular mutational screenings and the prognostic role of molecular alterations [69] should be considered in the evaluation of SCT. Recent reviews [67,68] and consensus treatment recommendations (Table 3) confirm that allogeneic SCT confers survival benefits in patients with HR-MDS compared to non-transplantation approaches. The use of allogeneic SCT is increasing in older patients with acceptable performance status and without a severe burden of comorbidities [66]. Administration of novel therapies before or after transplantation may decrease the risk of disease relapse in selected populations [68]. Consensus recommendations on indications, conditioning regimens, and donor selection for allogeneic SCT have facilitated the standardization of clinical practices in this setting [68]. Less intense conditioning regimens and improvements in supportive therapy have reduced transplant-related mortality and increased access to this curative treatment. The timing of the transplant procedure in the course of the disease is crucial, and factors such as disease progression, geriatric assessment, comorbidities evaluation, and the identification of transplant-specific risk factors should be considered [66]. Allogeneic SCT remains the only curative option for MDS and is also used to treat MDS-associated rheumatologic conditions, such as VEXAS syndrome [70].

## 10. Summary and Conclusions

Recent years have witnessed significant advancements in the understanding and treatment of MDS. Updated classifications [1,2] incorporating genomic data and the development of prognostic scoring systems like IPSS-M [42] have improved risk stratification, paving the way for personalized treatment approaches and identified therapeutic targets [5,62]. From a practical point of view, new prognostic systems incorporating molecular data have moved a significant proportion of patients into higher risk categories of disease. Furthermore, the innovations introduced by the new classifications [1,2], as well as the availability of novel target drugs and more tolerable and safe formulations of intensive chemotherapy, such as CPX-351 [113], are progressively shifting haematological clinical practice, including treating HR-MDS similarly to AML [2,117]. Therefore, progress in molecular genomic knowledge and therapeutic advances are determining a progressive cultural transformation of haematological clinical practice. Furthermore, there remain problematic delays in the regulatory approval of new drugs [117] and disparities in the availability between different continents and, even, between their different nations. The MDS community’s commitment to overcoming these disparities should be increasingly urgent and pressing as new effective drugs are tested in clinical trials and should be available for clinical use in daily haematological practice. With these premises in mind, we should recognize that changes in the MDS scenario have been remarkable. Indeed, in recent years, expanding treatment options in MDS have included some important novel therapies for anaemia in RS-MDS, like luspatercept, which has proven to be more effective than EPO in LR-MDS patients and those affected by SF3B1-mutant MDS [97]. Therefore, it could be expected that luspatercept may soon replace the traditionally used ESAs in daily clinical practice [118]. Furthermore, other novel and alternative agents to relieve anaemia, such as imetelstat [98], are in the advanced phase of their development for anaemic patients with LR-MDS. In such a setting, the approval of eltrombopag for thrombocytopenic patients, in line with the recently published results of the phase III-controlled trial [85], will provide meaningful and substantial progress in this difficult-to-treat setting. Again, ongoing research on venetoclax and other emerging agents, such as oral HMAs as the potential backbone for several treatment combinations [119,120], is shaping the landscape of MDS management, offering potential useful therapeutic alternatives for HR patients, which continue to represent an unmet clinical need. Indeed, the safety and activity of the first oral combination of decitabine plus cedazuridine and venetoclax in patients with HR-MDS has been reported [120] with an ORR of 95%. Therefore, this entirely oral treatment combination, other than providing highly effective therapeutic activity, might be paradigm-changing for HR-MDS patients, avoiding the inconvenient parenteral administration route, reducing travel times, and providing other obvious subjective benefits for family and caregivers [119,120]. In addition, attempts to modulate the immune system and inflammatory pathways, inducing defects in HSC progenitor cells, may lead to the development of immunotherapeutic interventions, such as CAR-T, vaccines, and immune checkpoint inhibitors, mitigating immunological dysregulation and altered inflammation in MDS [8,121]. In particular, combining the specific targeting of transcriptional regulators with immune checkpoint inhibitors may successfully implement the treatment of fitted MDS patients. This implies identifying and prospectively validating composite biomarkers as a way of personalizing patient management [8].

Notably, advancements in supportive care and QoL-preserving strategies can contribute to the holistic well-being of MDS patients [64]. In conclusion, the evolving understanding of MDS pathogenesis, coupled with emerging therapeutic options, holds promise for improved patient outcomes. Further research and clinical trials will continue to refine diagnostic criteria, prognostic systems, and treatment strategies, ultimately providing more tailored and effective care for individuals with MDS [122].

## Figures and Tables

**Figure 1 cancers-16-01563-f001:**
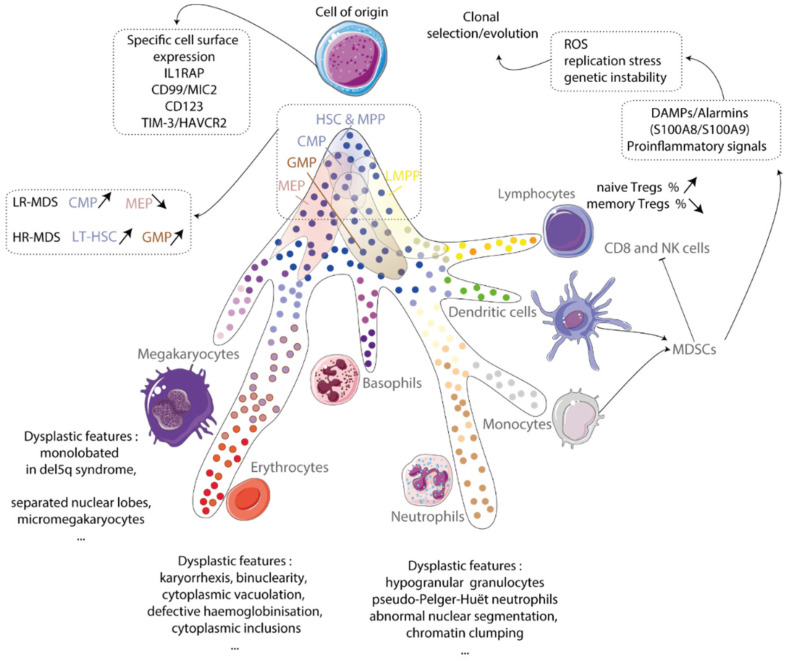
Cell of origin of myelodysplastic syndromes (MDSs). MDS originates from clonal hematopoietic stem cells (HSCs). This results in the production of abnormal megakaryocytes, erythroblasts, and granulocytes. An inflammatory environment made up of mostly myeloid-derived suppressive cells (MDSCs) contributes to the generation of reactive oxygen species (ROS), replication stress, and genetic instability, leading to the development of clonal evolution. IL-1 receptor accessory protein and TIM-3/HAVCR2 are important proteins involved in this process. The hematopoietic stem cell has two types: long-term hematopoietic stem cells (LT-HSCs) and multipotent progenitors (MPPs). The common myeloid progenitor (CMP), granulocytic myeloid progenitor (GMP), and megakaryocytic-erythroid progenitor (MEP) are all progenitor cells that can be involved in MDS development. Regulator T-cells (Tregs), natural killer cells (NKs), and damage-associated molecular patterns (DAMPs) are also involved in MDS. Taken from [8].

**Figure 2 cancers-16-01563-f002:**
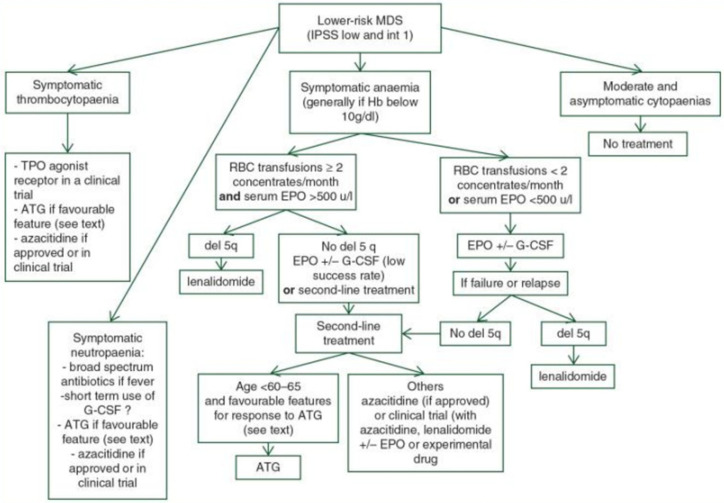
Treatment algorithm for lower-risk MDS Legend: ATG (anti-thymocyte globulin), EPO (erythropoietin), G-CSF (granulocyte colony-stimulating factor), Hb (haemoglobin), IPSS-R (revised International Prognostic Scoring System), MDS (myelodysplastic syndromes), MDS-RS (myelodysplastic syndrome with ring sideroblast), RBC (red blood cell), and TPO-RA (thrombopoietin receptor agonist). For intermediate-risk MDS patients according to the IPSS-R, whether they should undergo treatment for lower-risk MDS or higher-risk MDS depends on several other factors. These include age, comorbidities, importance of cytopenias, somatic mutations, and the effect of first-line treatment (taken and adapted from source [62]).

**Figure 3 cancers-16-01563-f003:**
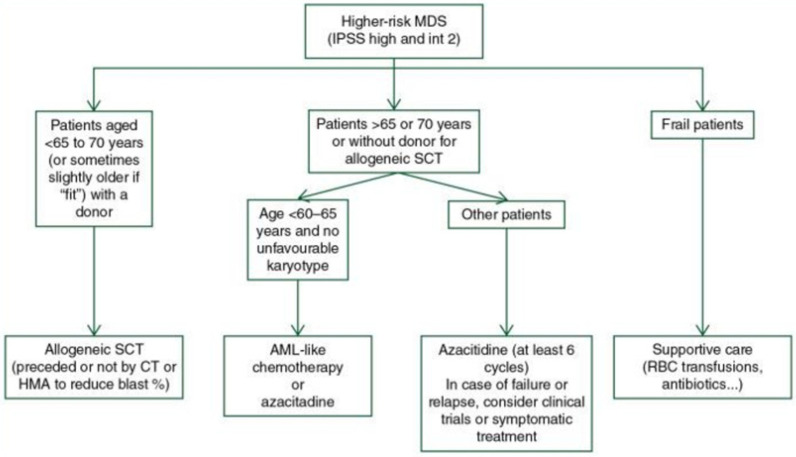
Treatment algorithm for high-risk MDS [62]. The legend explains that allogeneic stem cell transplantation (Allo-SCT), chemotherapy (ChT), and hypomethylating agents (HMAs) are the possible treatments for MDS. Acute myeloid leukaemia (AML) and red blood cells (RBCs) are also mentioned. It is worth noting that for IPSS-R intermediate-risk MDS patients, deciding whether to initially receive treatment for low-risk MDS or high-risk MDS depends on other factors such as age, comorbidities, the importance of cytopenia, somatic mutations, and the effect of first-line treatment (taken and adapted from source [62]).

**Table 1 cancers-16-01563-t001:** Comparison of the 5th edition WHO [1] and 2022 ICC [2] classifications of MDS *.

BM blasts	5th edition WHO ^	ICC 2022 °	Notations and comments ^ Myelodysplastic syndromes” are termed “myelodysplastic neoplasms”
<5%	MDS low blasts	MDS-SLD MDS-MLD	ICC 2022 includes single (>10% for one lineage) vs. multilineage dysplasia (>10% for more than one lineage).
	MDS-RS	MDS-RS	° also in the absence of SF3B1mutation
	MDS isolated del (5q)	MDS isolated del (5q)	
	Biallelic TP53 inactivation		
5–9%	MDS IB1	MDS excess of blasts	^ “increased” instead of “excess” of blasts
	MDS-f		^ BMF grade 2 or 3 and BM blasts >5%
10–19%	MDS IB2		° MDS/AML
20%	AML	AML	both classifications adopt this blast cut-off to distinguish MDS from AML.

* Adapted from refs [1,2]. Abbreviations: BM: bone marrow; MDS: myelodysplastic syndrome/neoplasm; SLD: single-lineage dysplasia; MLD: multilineage dysplasia; IB1: increased blast-1; IB2: increased blast-2; f: fibrosis; BMF: bone marrow fibrosis; AML: acute myeloid leukaemia. ^: WHO; °: ICC.

**Table 2 cancers-16-01563-t002:** Revised International Prognostic Scoring System (IPSS-R) ° for MDS [56].

Prognostic Characteristics	Points						
	0	0.5	1	1.5	2	3	4
Cytogenetic risk category *	Very good		Good		Intermediate	Poor	Very poor
Blasts in bone marrow, %	<2		>2–5%		5–10%	10%	
Haemoglobin, g/dL	≥10		8–<10	<8			
Platelet count, ×10^9^	≥100	50–<100	<50				
Absolute neutrophil count, ×10^9^	≥0.8	<0.8					
IPSS-R risk group	Score	Median OS (years)			Median time to 25% AML evolution (years)		
Very low	≤1.5	8.8			NR		
Low	>1.5–3	5.3			9.4		
Intermediate	>3–4.5	3.0			2.5		
High	>4.5–6	1.6			1.7		
Very high	>6	0.8			0.7		

° Sum scores on a 0–10-point scale; * cytogenetic risk group, very good: -Y, del(11q); good: normal; del(5q) ± 1 other abnormality del(20q), or del(12p); intermediate: +8, (17q), del(7q), +19, any other abnormality not listed including the preceding with one other abnormality; poor: −7 ± del(7q), inv(3)/t(3q)/del(3q), any three separate abnormalities; very poor: more than three abnormalities, especially if 17p is deleted or rearranged. OS: overall survival; AML: acute myeloid leukaemia; NR: unreached.

**Table 3 cancers-16-01563-t003:** Recommendations for allogeneic SCT in MDS patients (Adapted from [68]).

Transplantation Indications
All MDS patients should be considered for allogeneic SCT. Patients with HR-MDS should be referred at diagnosis or early in the disease course; for those with LR, the referral is less urgent, although they should be closely observed and quickly referred as appropriate.
Patient-specific considerations
Eligibility for SCT should not be limited by age nor comorbidities, but the performance status and the severity of the comorbid conditions significantly influence transplant success.
Disease considerations
Allogeneic SCT should be considered early for t-MDS and HR-MDS. Germline mutations should be considered in potential related donors.
Disease-directed therapy and transfusion overload management
Patients may receive disease-directed therapy before SCT, although its value is uncertain. Pre-transplant iron overload should be managed with iron chelation therapy.
Conditioning regimen
Both RIC and MAC regimens are acceptable for allogeneic SCT in MDS. MAC may be preferred in fit LR patients (lower post-transplant relapse rate).
Alternative donors
If an HLA-matched donor is not available, haploidentical relatives, mismatched unrelated donors, and umbilical cord blood can be considered as alternative options.
Post-transplantation
Maintenance therapy after SCT has not been firmly established by clinical trials. For relapsed disease after SCT, various treatment options including IST, novel agents, DLI, or a second SCT should be considered.

Legend. DLI: donor lymphocyte infusion; IST: immunosuppressive therapy; MAC: myeloablative conditioning; RIC; reduced-intensity conditioning.

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
