# Peer review of "Latest Insights and Therapeutic Advances in Myelodysplastic Neoplasms"

_cancers, 2024, doi:10.3390/cancers16081563_

Round 1

Reviewer 1 Report

Comments and Suggestions for Authors

This manuscript aims to review the latest conceptual and therapeutic advances in adult MDS. However, there are quite a few recent review articles on this topic. There are some errors in the manuscript as shown in bold text.

Regarding the section of Abstract:

Lines 30-31: The term of “bone marrow (BM) failure” is redundant because it has the same meaning of “ineffective hematopoiesis” in the same sentence. Myelodysplasia and ineffective hematopoiesis are the two hallmarks for MDS, and BM failure may have different meanings to different people. 

Regarding the section of Disease overview and pathogenesis:

Line 58 “associated complications”: The authors need to be specific about the complications here.

Lines 62-65: This sentence is wordy and difficult to read. Please rephrase it.

Lines 83-91: The statement on the prognosis of de novo MDS vs. therapy-related MDS (t-MDS) including t-AML is incorrect.  I would recommend reading and citing the following seminal papers, which confirm a shorter median survival for t-MDS.

Smith SM, Le Beau MM, Huo D, Karrison T, Sobecks RM, Anastasi J, et al. Clinical-cytogenetic associations in 306 patients with therapy-related myelodysplasia and myeloid leukemia: the University of Chicago series. Blood. 2003;102:43–52.

Larson RA. Therapy-related myeloid neoplasms. Haematologica. 2009 Apr;94(4):454-9. doi: 10.3324/haematol.2008.005157. PMID: 19336749; PMCID: PMC2663607.

Zeidan AM, Al Ali N, Barnard J, Padron E, Lancet JE, Sekeres MA, et al. Comparison of clinical outcomes and prognostic utility of risk stratification tools in patients with therapy-related vs de novo myelodysplastic syndromes: a report on behalf of the MDS Clinical Research Consortium. Leukemia. 2017;31:1391–7.

Moreno Berggren, D., Garelius, H., Willner Hjelm, P. et al. Therapy-related MDS dissected based on primary disease and treatment—a nationwide perspective. Leukemia 37, 1103–1112 (2023). https://doi.org/10.1038/s41375-023-01864-6

Regarding the section of Diagnosis and classifications:

Lines 97-98: The word “traditionally” is incorrect here because BM cellularity and reticulin fibrosis degree are new items added to the WHO-5 and the 2022 ICC. They were not included in the previous editions. (There were debates about the clinical significance of BM cellularity and reticulin fibrosis in the past.)

Lines 99-104: This sentence is too long and difficult to read. Please rephrase it.

Regarding the Table 1: The WHO-5 does not include single vs. multilineage dysplasia as does the 2022 ICC. This difference should be included in the table.

Regarding the section of Prognostic Systems and Risk-Stratification:

Lines 152: The word “high-risk” is missing in the sentence.

Lines 169-174: This very long sentence is grammatically correct. Please rephrase it.

Regarding the section of Clinical management:

Line 192: Missing the word MDS here.

Line 194: What is “h-MDS”?

Regarding the section Higher-risk MDS:

Why did the authors use all bold text for this section?

The format of Table 2 looks odd to my eyes and is also difficult to follow.

Comments on the Quality of English Language

There are some grammatical errors and typos in the manuscript. At least moderate editing of English language is required.

Author Response

Thank you! We have revised the paper, at our best, according to your suggestions. 

Reviewer 2 Report

Comments and Suggestions for Authors

This review is about myelodysplastic syndromes (MDS). Myelodysplastic syndromes (MDS) comprise a group of hematologic malignancies characterized by clonal hematopoiesis, one or more cytopenias (ie, anemia, neutropenia, and/or thrombocytopenia), and abnormal cellular maturation. Patients are at risk for symptoms related to anemia, infection, and bleeding, and they have variable rates of transformation to acute myeloid leukemia (AML).

This review is well written, and it is easy to read and understand. However, the different sections and paragraphs are large, and it is quite difficult to identify the most relevant and key points, which could be highlighted using figures or tables.

Specific comemnts:

(1) Line 55. Please write all gene names in italics.

(2) Line 80. Regarding CART and t-MDS. How long does it take to develop MDS after CART?

(3) I am aware that the abbreviations are shown in the footnotes of the table. But may you please reduce the quantity of abbreviations in Table 1. For example, the reader may benefit from EB to be written as excess of blasts.

(4) Regarding lines 135-147. As I understand, there are similarities and differences between the WHO5 and ICC2022. Why two different approaches have been defined? Is there any fundamental difference in the understanding of the pathological features of MDS? What are the clinical implications? Would this lead to different treatment of the patients?

(5) May you please expand, or make a section, regarding the pathological mechanisms of MDS, including tables and/or figures? What are the genomic changes? What are the immune microenvironmental changes? What molecules are affected by drugs or chemicals? A summarizing figure would be very informative to the readers.

(6) Immunohistochemistry can be useful for identifying the cellular lineage and/or the aberrations of the cellular maturation. Could you please add this information? For example:

Erythroid precursor cells – Staining for glycophorin (CD235a), transferrin receptor (CD71), and/or GATA1 can aid in detecting erythroid precursor cells.

Blasts – Staining for CD34, CD117, CD33, myeloperoxidase, and lysozyme can assist in quantifying blasts and myeloid progenitors.

Megakaryocyte dysplasia – Staining for CD41 and/or CD61 can aid in detection of dysplastic or immature megakaryocytes.

Lineage infidelity – Staining for myeloid and lymphoid markers can help to detect lineage infidelity, confirm the presence of bi- or tri-lineage dysplasia, and exclude a lymphoid origin of primitive blasts.

(7) Section LR-MDS is one paragraph long. From lines 202 to 285. It is a matter of style, but this is very hard to read. Could you please add a table summarizing the most relevant findings?

(8) Could you please add in appendix the Revised international prognostic scoring system (IPSS-R) in myelodysplastic syndrome?

(9) Why is del(5q) so important? What are the relevant genes in that location?

(10) Please also add table/figure summarizing the content of HR-MDS.

(11) As I understand, this review focused on adult. Should the MDS in children be mentioned as well?

Author Response

Thank you! We have revised, at our best, the paper according to your suggestions. 

Reviewer 3 Report

Comments and Suggestions for Authors

I have the following concerns:

1. Please separate the main text of your manuscript to many paragraphs. You need 5 or 6 paragraphs for every section, so that the text will not be tiring for the reader. 

2. Have the novel classifications (WHO 2022 and ICC) plus the IPSS-M altered our treatment guidelines towards MDS? If yes, in what way? Please add a paragraph. 

3. Which are the most common mutations seen in a. low risk and b. high risk MDS?

4. Which is the minimal diagnostic gene panel that should be given to every MDS patient for a proper evaluation and treament decision?

5. Do you believe that HMAs should be given in low-risk MDS patients? Please add a novel paragraph.

6. Based on your clinical experience, are there MDS patients responding to luspatercept without having the SF3B1 mutation? Are there possible other mutations that respond to luspatercept?

7. Are there any data regarding the use of lenalidomide plus luspatercept in MDS patients? Either concomitant or one after the other? Are there any known clinical trials?

8. Please add a table highlighting tratment of a. low MDS and b. high risk MDS 

Author Response

(The authors gave the same response as above.)

Round 2

Reviewer 1 Report

Comments and Suggestions for Authors

All my comments and suggestions have been addressed and adopted by the authors. However, I did not see the figures 1,2 and 3? 

Author Response

Thank you. Done

Reviewer 3 Report

Comments and Suggestions for Authors

I have no further concerns. 

Author Response

Thank you!